# Spectral-Aware Sparse Communication and Entropy-Balanced Tasking in Multi-Agent Systems

## Abstract

Multi-agent systems (MAS) face scalability constraints stemming from dense messaging—raising bandwidth and energy—and from imbalanced tasking that produces bottlenecks, especially under non-stationary, LLM-driven workloads. We introduce a unified framework that prunes redundant links using an information-theoretic priority and enforces connectivity via a spectral $\lambda_2$ guard, and balances workload with an entropy-regularized assignment under capacity constraints; an MDP allocator adapts thresholds and repairs to system drift. We prove that the $\lambda_2$-guarded repair preserves connectivity and, under standard spectral envelope assumptions, the sparsified graph approximates dense-graph dynamics; we analyze discrete-time stability via Jury/Nyquist bounds. Across GSM8K, MMLU, and SMACv2, the method improves over a dense complete-graph baseline by +6.12%/+5.59%/+4.76%, reduces active links by 28%, and shows the strongest robustness under 30% edge drops – all while keeping per-iteration communication proportional to the number of active edges. These results indicate a practical route to communication-efficient, entropy-balanced coordination for LLM-augmented MAS and cooperative control.

## 1 Introduction

Multi-agent systems (MAS) (van der Hoek & Wooldridge, 2008; Dorri et al., 2018; McArthur et al., 2007a;b) have become a critical paradigm for solving complex, distributed tasks across various domains, including robotics, sensor networks, and autonomous vehicles (Groenewald et al., 2024; Harshbarger et al., 2024). With the rise of large language models (LLMs) (Guo et al., 2024; Dong, 2024), MAS capabilities have expanded, enabling agents to generate objectives, adapt to dynamic environments, and make autonomous decisions. Recent advancements, such as Auto-GPT (Torantulino, 2023) and BabyAGI (Nakajima, 2023), have demonstrated the potential of LLM-powered MAS for collaborative task solving, where specialized agents coordinate towards shared goals (Chen et al., 2024; Händler, 2023).

However, as MAS scales, several challenges emerge, including high communication overhead, inefficient task allocation, and suboptimal resource distribution (Lee et al., 1998; Turner & Jennings, 2000). Specifically, large-scale MAS often face issues with redundant communication (Hu et al., 2023; Sun et al., 2024b), leading to network congestion and resource inefficiencies. Moreover, task misalignment due to agent heterogeneity further exacerbates these challenges (Wilson, 2022; Mallampati & Ravichandar, 2023), resulting in inefficiencies in workload distribution.

These challenges are further compounded in LLM-powered MAS. Unlike traditional MAS, where tasks are predefined and agent roles remain static, LLM-driven agents dynamically generate tasks and adapt their reasoning in real time (Yang et al., 2024). This increased task uncertainty introduces greater complexity in dependency management and computational load balancing. Standard heuristic or gradient-based optimization techniques struggle to efficiently distribute tasks in such high-dimensional environments (Parker, 2013; Shi & Yu, 2023), necessitating more sophisticated coordination mechanisms.

To address these challenges, we propose a novel framework that integrates three key innovations: (1) sparsity-aware communication optimization; (2) entropy-guided task decomposition; and (3)

MDP-based adaptive task allocation. Our framework specifically targets three critical issues that commonly arise in large-scale MAS. First, to curb the quadratic cost of fully connected messaging (Charalambous et al., 2025; Krishnan, 2025), the communication module suppresses redundant exchanges while maintaining connectivity and convergence properties. Second, to mitigate workload skew in heterogeneous agents (Alirezazadeh & Alexandre, 2024), the task module maximizes assignment entropy under capacity constraints, yielding balanced utilization. Third, to handle non-stationary, LLM-driven tasks (Choudhury et al., 2022; KyeG, 2024), the allocation module casts coordination as a sequential decision problem, adapting decisions to the evolving system state. Together, these components enhance scalability, balance, and adaptability in modern MAS.

To evaluate the framework, we conduct extensive experiments on GSM8K (Cobbe et al., 2021), MMLU (Hendrycks et al., 2021) and SMACv2 (Ellis et al., 2023) against dense-graph, pruning-based, MARL, and GNN baselines. The method attains 95.86% accuracy on GSM8K, 91.35% on MMLU and 40.30% on SMACv2, exceeding the Complete-Graph (West, 2001) baseline by +6.12%, +5.59% and 4.76% respectively, while cutting redundant pairwise exchanges by 28% without degrading connectivity or accuracy.

Our main contributions can be summarized as follows:

1. We propose a sparsity-aware communication strategy that optimizes network topology using Laplacian eigenvalue analysis, reducing communication overhead while maintaining system convergence.
2. We introduce an entropy-based task decomposition mechanism that dynamically balances task distribution by maximizing Shannon entropy, ensuring a more efficient allocation of resources across heterogeneous agents.
3. We develop an MDP-driven adaptive task allocation framework, which models task allocation under uncertainty and adapts dynamically to evolving task dependencies.
4. On GSM8K, MMLU and SMACv2 the framework achieves 95.86%, 91.35% and 40.30% accuracy and win-rate, respectively; improves over a Complete-Graph baseline by +6.12%/ + 5.59%/ + 4.76%; reduces redundant communication by 28%;

This paper is structured as follows: In Section 2, we provide the necessary preliminaries and definitions, grounding our work in existing literature. In Section 3, we present the detailed methodology, explaining our communication strategy, entropy-based task decomposition, and MDP-based task allocation. Section 4 outlines our experimental setup, performance evaluation, and comparison with state-of-the-art methods. Finally, Section 5 concludes with a discussion of results and future work.

## 2 PRELIMINARIES AND DEFINITIONS

We model a multi-agent system (MAS) as a connected, weighted, undirected graph $G = (V, E, W)$ with Laplacian $L = D - A$. Throughout the main text we work with the canonical first-order consensus abstraction

$$\dot{x}(t) = -L\,x(t), \tag{1}$$

which cleanly links communication topology to convergence; second-order models standard in robotics are deferred to Appendix A, where we state analogous stability conditions (Dibaji & Ishii, 2017; Cheng et al., 2019); complementary sampled-data stability bounds are given in Appendix B.

### 2.1 SPECTRAL QUANTITIES AND CONVERGENCE

Let $0 = \lambda_1(L) < \lambda_2(L) \leq \cdots \leq \lambda_n(L)$ be the eigenvalues of $L$. For connected graphs, equation 1 reaches consensus with a rate governed by the algebraic connectivity $\lambda_2(L)$; larger $\lambda_2(L)$ implies faster mixing and improved robustness to perturbations (Lin et al., 2014; Franceschelli et al., 2013). We will use $\lambda_2(L)$ as a spectral guard when sparsifying communication.

### 2.2 SPECTRAL SPARSIFICATION

To reduce messaging while preserving convergence behavior, we replace $L$ by a sparser Laplacian $L_S$ that approximates $L$ in quadratic form:

$$(1 - \varepsilon)\,L \;\preceq\; L_S \;\preceq\; (1 + \varepsilon)\,L, \tag{2}$$

with a target connectivity constraint $\lambda_2(L_S) \geq \lambda_{\text{target}}$. Condition equation 2 controls distortion of energies $x^\top L x$ and thereby maintains the qualitative dynamics of equation 1 while reducing the number of active links (Chen et al., 2023; Sadhanala et al., 2016). The resulting communication budget is quantified by $S_d$ (defined in Section 3).

### 2.3 ENTROPY-BASED TASK DECOMPOSITION

We encode assignments in $P \in [0,1]^{m \times n}$ (agents × tasks), with capacity constraints $\sum_j P_{ij} \leq c_i$ and per-task normalization $\sum_i P_{ij} = 1$. To mitigate workload skew in heterogeneous MAS, we promote diversity/balance via entropy:

$$\max_P \quad \sum_{i=1}^m H(P_{i:}) + \lambda \sum_{j=1}^n H(P_{:j}) \quad \text{s.t.} \quad \sum_j P_{ij} \leq c_i, \quad \sum_i P_{ij} = 1, \tag{3}$$

where $H(q) = -\sum_k q_k \log q_k$. This objective yields capacity-aware, balanced allocations used by our scheduler in Section 3.

### 2.4 ADAPTIVE ALLOCATION AS A SEQUENTIAL DECISION PROBLEM

Non-stationary, LLM-driven tasks require adaptation. We cast coordination as an MDP $(\mathcal{S}, \mathcal{A}, P, R)$, where states summarize load and graph statistics, actions adjust communication masks and task weights, and rewards trade task performance against messaging cost (BELLMAN, 1957). Learning details and algorithmic variants are given in Appendix D.

## 3 METHODOLOGY

While second-order dynamics effectively model agent interactions and ensure consensus, their reliance on dense communication networks and unstructured task allocations introduces inefficiencies in scalability, energy consumption, and convergence time. Specifically, dense communication leads to quadratic growth in message complexity, while imbalanced task decomposition results in redundant computations and agent overload (Gao et al., 2023). These challenges have been well-documented in the literature, particularly with respect to large-scale coordination in multi-agent environments (Lee et al., 1998; Sun et al., 2024a).

To overcome these challenges, we propose a unified framework that integrates sparsity-aware communication optimization and entropy-driven task decomposition. The design goal is to minimize active links subject to a connectivity guard and to rebalance tasks online. By selectively pruning redundant communication links and dynamically balancing workload allocations, our framework reduces computational and communication overhead while preserving network connectivity, system scalability, and robust convergence guarantees. Our approach builds upon foundational work in multi-agent coordination and communication sparsification (Spielman & Teng, 2011; Ahn et al., 2013), introducing novel entropy-guided task allocation techniques for dynamic task environments.

### 3.1 OPTIMIZED COMMUNICATION THROUGH SPARSITY

#### 3.1.1 COMMUNICATION GRAPH AND SPARSITY

In a multi-agent system, agents interact via a communication graph $G = (V, E, A)$, where: $V = \{1, 2, \ldots, m\}$ represents the set of $m$ agents (nodes); $E \subseteq V \times V$ is the set of undirected communication links (edges), where $(i, j) \in E$ implies agent $j$ can receive information from agent $i$; $A = [a_{ij}]$ is the adjacency matrix, where $a_{ij} > 0$ if $(i, j) \in E$, and $a_{ij} = 0$ otherwise.

The Laplacian matrix $L = D - A$ is a fundamental mathematical tool for describing the structure of $G$, where $D$ is the degree matrix and $A$ is the adjacency matrix. Specifically:

$$L_{ii} = \sum_{j \in N_i} a_{ij}, \quad L_{ij} = -a_{ij} \, (i \neq j). \tag{4}$$

Dense communication graphs, where a large number of links are active, exacerbate the computational and communication costs. Specifically, in a fully connected graph, the number of edges scales

as $|E| = m(m-1)/2$, which grows quadratically with the number of agents $m$. The presence of many redundant links increases the overhead for maintaining and synchronizing information.

To address these challenges, sparsity is introduced by selectively retaining only essential communication links, effectively reducing the number of active edges $|E|$. Let $E_{\text{full}}$ denote the edge set of the fully connected graph such that $|E_{\text{full}}| = m(m-1)/2$. The sparsity ratio $S_d$ quantifies the fraction of retained links:

$$S_d = \frac{|E|}{|E_{\text{full}}|}. \tag{5}$$

In what follows we measure pre-iteration communication by the number of edge messages $\Theta(|E_{\text{active}}|)$ and treat latency separately via graph metrics such as $D(G)$.

By enforcing connectivity via a $\lambda_2$-guarded repair step, even with fewer edges, sparsity minimizes overhead while preserving the key properties necessary for convergence and robustness. This approach draws from prior work in communication graph sparsification, where the focus is on reducing complexity while ensuring system stability.

### 3.1.2   COMMUNICATION PRIORITY MATRIX

To prioritize critical communication links and dynamically identify the most informative communication links, we define the *communication priority matrix*:

$$C_{ij} = \frac{1}{1 + I(X_i; X_j)}, \tag{6}$$

where $I(X_i; X_j)$ denotes the mutual information between agents $i$ and $j$. The matrix quantifies the relevance of communication between pairs of agents, with lower mutual information resulting in higher priority values. Links are retained if $C_{ij} > \tau$, where $\tau > 0$ is a sparsity threshold. This ensures that communication links are preserved only when they significantly contribute to system coordination and information exchange.

The sparsity threshold is adaptively updated based on the real-time network state:

$$\tau_{t+1} = \tau_t + \kappa \cdot \left( \frac{\lambda_2(L_S) - \lambda_{\text{target}}}{\lambda_{\text{target}}} \right), \tag{7}$$

where $\lambda_{\text{target}}$ is the desired network connectivity level, and $\kappa$ is an adaptation rate.

The communication priority matrix directly addresses the challenge of redundant communication by dynamically identifying and prioritizing links that provide the most informative updates. By focusing on critical links, this approach reduces the overall communication overhead, conserves bandwidth, and ensures that agents exchange only the most relevant information, thereby accelerating convergence. This method extends the work on adaptive communication strategies by (Sun et al., 2024a), who highlighted the role of information-theoretic methods in optimizing network performance.

### 3.1.3   SPARSITY-ADJUSTED DYNAMICS

Given the pruned and repaired graph $G_S$, we analyze the closed-loop dynamics under its Laplacian $L_S$ and state the resulting stability and convergence guarantees. For sampled-data updates with period $T$, we adopt the Nyquist-style sufficient bound of Appendix B. The sparsity-adjusted Laplacian $L_S$ governs the system's dynamics:

$$\frac{dx(t)}{dt} = -L_S x(t), \tag{8}$$

where $L_S$ is derived from the original Laplacian matrix $L$ after applying the communication priority matrix to retain only the most relevant links. The convergence behavior of the system is then determined by the adjusted algebraic connectivity $\lambda_2(L_S)$:

$$\|x(t) - x^*\| \leq \|x(0) - x^*\| e^{-\lambda_2(L_S)t}. \tag{9}$$

A well-connected sparse graph ensures that $\lambda_2(L_S)$ remains sufficiently large to guarantee efficient consensus across the agents (Li et al., 2018).

Adjusting the dynamics with sparsity ensures that the system retains its ability to converge while significantly reducing computational and communication requirements. By pruning less critical links, the sparsity-adjusted dynamics maintain the essential properties of the original system without incurring the high costs of dense communication graphs.

The communication priority matrix directly informs the construction of $L_S$, as the retained links are determined based on their priority values. This establishes a clear pipeline where communication prioritization translates into optimized dynamics, ensuring that sparsity enhances the system without compromising its performance.

### 3.1.4 COMPLEXITY REDUCTION

Introducing sparsity reduces the pre-iteration message count to $C_{\text{comm}} = \Theta(|E_{\text{active}}|)$, where $|E_{\text{active}}| = |E_S|$. For a dense baseline $|E| \approx \frac{m(m-1)}{2}$, this yields $C_{\text{comm}} = \Theta(S_d \cdot m^2)$, under a degree cap $k$, $C_{\text{comm}} = \Theta(km)$. Pre-round dissemination latency scales separately with the graph diameter $D(G)$. See Appendix D.6 for a full accounting.

Reducing communication complexity not only lowers the bandwidth and energy requirements but also enables the system to scale to a larger number of agents. This improvement is particularly significant in resource-constrained environments where minimizing energy consumption is critical.

The complexity reduction arises naturally from the integration of the communication priority matrix and sparsity-adjusted dynamics. The priority matrix selects the most relevant links, while the sparsity-adjusted Laplacian ensures these selected links maintain system connectivity. Together, these components achieve an optimal balance between performance and efficiency, making the overall framework scalable and robust.

## 3.2 ENTROPY-BASED TASK DECOMPOSITION

Entropy provides a natural measure of uncertainty, enabling dynamic and balanced task allocation. The entropy of a task allocation $X$ is: $H(X) = -\sum_{i=1}^{n} p_i \log(p_i)$, where $p_i$ represents the probability of assigning task $i$. Maximizing $H(X)$ ensures even workload distribution across agents.

By pruning edges using a Shannon-entropy criterion – retaining links whose incremental information exceeds an adaptive threshold – we preserve the giant component and maintain algebraic connectivity: the sparsified Laplacian $L_S$ satisfies $(1 - \epsilon)L \preceq L_S \preceq (1 + \epsilon)L$ and $\lambda_2(L_S) \geq \lambda_{\text{target}} > 0$. Additionally, consensus over $L_S$ converges linearly with rate $1 - \alpha\lambda_2(L_S)$ for $\alpha \in (0, 2/\lambda_{\max}(L_S))$, and the degradation in rate/stability relative to the dense graph is $O(\epsilon)$.

To formally justify entropy-based sparsification, we prove in Appendix C that pruning edges based on Shannon entropy preserves the largest connected component of the communication graph while reducing the number of edges to $O(m \log m)$. Additionally, we establish a direct connection between entropy-guided sparsity and mutual information minimization, ensuring that only non-redundant communication channels are retained. Finally, we show that our method satisfies spectral sparsification bounds, guaranteeing that essential multi-agent coordination properties are preserved.

## 3.3 MDP-BASED TASK ALLOCATION

To enhance the effectiveness of task allocation in multi-agent systems, we model the process as a Markov Decision Process (MDP) (BELLMAN, 1957). This framework works in conjunction with the previously introduced communication prioritization and sparsity mechanisms. By integrating sparsity-adjusted dynamics and entropy-based task decomposition into the MDP framework, we create a system that dynamically adapts task allocation to optimize performance under uncertainty.

The system initializes the communication graph based on the communication priority matrix, retaining only the most critical links to form the sparsity-adjusted Laplacian matrix $L_S$. Using entropy-based task decomposition, initial task allocations are made to balance workloads across agents. The MDP framework refines these allocations over time, dynamically adapting to changing conditions such as incomplete tasks, agent availability, and communication updates.

The task allocation process is optimized using reinforcement learning methods (Gu et al., 2024), such as Q-learning (Watkins & Dayan, 1992). The following algorithm 1 outlines the process.

---

**Algorithm 1** Adaptive Sparse Communication & Entropy-Balanced Tasking

---

1: **Inputs:** target connectivity $\lambda_{\text{target}} > 0$; threshold adapt-rate $\kappa > 0$; initial threshold $\tau_0$; MI refresh period $K_{\text{MI}} \in \mathbb{N}$; repair budget $k_r \in \mathbb{N}$; assignment step size $\eta_P > 0$; feasible set $\mathcal{C}$ for $P$ (row/col/capacity constraints).

2: **Definitions:** redundancy score $r_{ij} := C_{ij} = \frac{1}{1+I(X_i;X_j)}$ (symmetrized); active undirected edge set $E_{\text{act}} \subseteq \{\{i,j\} : i < j\}$; Laplacian $L_S$ of $G_S = (V, E_{\text{act}})$.

3: **Output:** assignment matrix $P \in \mathcal{C}$; active links $E_{\text{act}}$; final threshold $\tau^\star$.

4: **Init:** estimate $I(X_i;X_j)$; set $r_{ij} \leftarrow C_{ij}$ (symmetrize); $E_{\text{act}} \leftarrow \{\{i,j\} : r_{ij} > \tau_0\}$; form $L_{S,0}$; $\widehat{\lambda}_2 \leftarrow \text{PowerIter}(L_{S,0})$; initialize $P_0 \in \mathcal{C}$ (e.g., uniform); set $\tau \leftarrow \tau_0, t \leftarrow 0$.

5: **while** $t < T_{\max}$ **do**

6:     (MI refresh) **if** $t \bmod K_{\text{MI}} = 0$ **then** re-estimate $I(X_i;X_j)$; update $r_{ij} \leftarrow C_{ij}$; **end if**

7:     (Threshold update) $\tau \leftarrow \tau + \kappa \cdot \dfrac{\widehat{\lambda}_2 - \lambda_{\text{target}}}{\lambda_{\text{target}}}$                     // Eq. (7)

8:     (Prune) $E_{\text{cand}} \leftarrow \{\{i,j\} : r_{ij} > \tau\}$; $E_{\text{act}} \leftarrow E_{\text{cand}}$; update $L_S$

9:     (Repair for $\lambda_2$)

10:     **while** $\text{PowerIter}(L_S) < \lambda_{\text{target}}$ **and** #added $< k_r$ **do**

11:         choose $e^\star \in E \setminus E_{\text{act}}$ by Fiedler-vector or effective-resistance heuristic

12:         $E_{\text{act}} \leftarrow E_{\text{act}} \cup \{e^\star\}$; update $L_S$; increment #added

13:     **end while**

14:     update $\widehat{\lambda}_2 \leftarrow \text{PowerIter}(L_S)$

15:     (Entropy-balanced assignment)

16:     $P \leftarrow \text{Proj}_{\mathcal{C}}\Big(P + \eta_P \nabla_P H(P)\Big)$, where $H(P) = -\sum_{i,j} P_{ij} \log P_{ij}$

17:     // (optionally weight row/col entropies; see App. D for exact gradients and $\mathcal{C}$)

18:     (Stopping) **if** $|\Delta\tau| \le \varepsilon_\tau$ **and** $\|P - P_{\text{prev}}\|_\infty \le \varepsilon_P$ for $T_{\text{hold}}$ steps **then break**

19:     $P_{\text{prev}} \leftarrow P$; $t \leftarrow t+1$

20: **end while**

21: **Return:** $P, E_{\text{act}}, \tau^\star \leftarrow \tau$

---

Implementation details (pseudocode, threshold update $\kappa$, MI refresh cadence $K_{\text{MI}}$, repair to enforce $\lambda_2(L_S) \ge \lambda_{\text{target}}$, and per-step complexity) are provided in Appendix D.

## 4 EXPERIMENTS

### 4.1 EXPERIMENTAL SETUP

To validate our framework, we evaluate across six datasets: **GSM8K**: 8.5k grade-school word problems for multi-step arithmetic reasoning (Cobbe et al., 2021); **MMLU**: 57 academic/professional subjects that test broad knowledge and reasoning (Hendrycks et al., 2021); **SVAMP**: a robustness-focused arithmetic benchmark created by perturbing math word problems to probe spurious shortcuts (Patel et al., 2021); **HumanEval**: functional code generation from docstrings, reported with pass@1 accuracy (Chen et al., 2021); **SMACv2**: a procedurally generated, partially observable multi-agent control suite reported as mean test win-rate (Ellis et al., 2023); **RWARE**: a cooperative multi-robot warehouse environment reported by episodic return (Papoudakis et al., 2020).

The performance is measured using three metrics: (i) **Accuracy** (LLM tasks: accuracy/pass@1; SMACv2: win-rate; RWARE: return); (ii) **Efficiency** via the edge ratio $S_d = |E_{\text{act}}|/|E|$ on an undirected graph, and (iii) **robustness** as relative decline under edge-drop $p = 0.3$; we choose the smallest $S_d$ that satisfies the spectral guard $\lambda_2(L_S) \ge \lambda_{\text{target}}$, yielding the conservative bound $S_d^{\min} \ge \lambda_{\text{target}}/\lambda_2(L)$.

We compare our method with the following baselines: **MAPPO** (Yu et al., 2022): A multi-agent reinforcement learning approach that uses centralized training and decentralized execution (CTDE) to optimize coordination. **GCN** (Kipf & Welling, 2017): A Graph Convolutional Network (GCN) that learns structured representations for agent communication, improving coordination. **Chain of Thought (CoT)** (Wei et al., 2023): Sequential reasoning using structured prompts. **Com-**

**plete Graph** (West, 2001): Fully connected communication among agents. **LLM-Blender** (Jiang et al., 2023): Coordination across agents via blending strategies. **AgentPrune-C, AgentPrune-L, AgentPrune-R**: Pruning strategies based on centrality, layer-level analysis, and random selection (Zhang et al., 2024). **Our Method**: Dynamic adaptive sparsity with entropy-based task decomposition, and MDP-driven task allocation.

## 4.2 RESULTS AND ANALYSIS

The results in Table 1 and Table 2 confirm that our method surpasses all baselines in **accuracy, communication efficiency, and robustness**, demonstrating its effectiveness in both arithmetic/language, complex reasoning tasks and multi-agent control. The inclusion of MARL-based, GNN-based, and classical optimization methods ensures a rigorous comparative assessment.

### 4.2.1 PERFORMANCE COMPARISON

Our method achieves **91.35% accuracy on MMLU** and **95.86% on GSM8K**, surpassing the Complete Graph baseline by **5.59% and 6.12%**, respectively. These gains stem from entropy-guided task decomposition, which diversifies competent contributors and reduces error correlation, and redundancy-aware sparsity, which filters low-information exchanges while preserving global connectivity. Among reference architectures, **MAPPO(89.20%/94.15%)** benefits from CTDE but lacks adaptive sparsity, and **GCN(87.94%/92.85%)** offers static mixing without task-aware allocation; both trail our approach on LLM reasoning. **LLM-Blender(92.45%,87.42%)** is competitive yet unconstrained, and **Complete Graph** incurs dense, non-selective messaging; pruning variants, **AgentPrune-C/L** improve over dense messaging but remain static, limiting their adaptability. On **SMACv2**, our method reaches **40.30% win-rate**, improving over the strongest communication baseline and complete graph by **3.5% and 4.76%**. On **RWARE**, we achieve a return of **35.40**, exceeding **LLM-Blender(+5.04) and Complete Graph(+5.44)**. These improvements align with our design: entropy-balanced assignment mitigates contention while adaptive sparsity with a $\lambda_2$ guard retains coordination under partial observability and task variability – yielding higher task scores without relying on dense, indiscriminate communication.

### 4.2.2 COMMUNICATION EFFICIENCY AND ROBUSTNESS

On **GSM8K**, our method uses the fewest links ($S_d = 0.62$), and on the harder **SMACv2** it remains the most efficient at $S_d = 0.68$. Competing settings require denser messaging, e.g., **MAPPO(0.68), GCN(0.72), LLM-Blender(0.75/0.83)**. The slight increase in $S_d$ on **SMACv2** reflects its stronger partial observability; our adaptive threshold with the $\lambda_2$ repair adds just enough edges to preserve connectivity while staying sparser than all baselines. Under edge-drop $p = 0.3$, our **robustness** decline is the lowest on both domains: **GSM8K(7.87%) and SMACv2(9.84%)**, compares with **LLM-Blender(9.62%/12.02%), Complete Graph(10.51%/13.14%), and CoT(11.23%/14.04%)**. These outcomes are consistent with the intended effect of entropy-balanced tasking and redundancy-aware sparsification.

### 4.2.3 SCALABILITY AND COMPLEXITY ANALYSIS

We validate that our sparsification method maintains essential communication links while reducing complexity to $O(m \log m)$. Theoretical results in C confirm that entropy-guided sparsity aligns with mutual information minimization and spectral sparsification, ensuring robust multi-agent coordination with minimal redundancy.

Although methods like LLM-Blender achieve high accuracy, they incur significant communication costs. In contrast, our method achieves sublinear communication complexity: $O(S_d \cdot m \log m)$ compared to the baseline Complete Graph's quadratic complexity $O(m^2)$.

While our framework demonstrates state-of-the-art performance in multi-agent coordination tasks, its scalability to large-scale, real-world MAS remains a critical consideration. First, our reliance on Laplacian eigenvalue analysis for optimizing communication introduces computational overhead, which may limit real-time feasibility in highly dynamic systems. Potential approximate spectral sparsification techniques could alleviate this issue. Additionally, the framework assumes homogeneous agents, whereas real-world MAS often consist of heterogeneous agents with varying compu-

Table 1: Performance comparison across multiple datasets. Best results are highlighted in bold.

| METHOD | MATH & LANGUAGE | | COMPLEX REASONING | | MULTI-AGENT CONTROL | |
|---|---|---|---|---|---|---|
| | MMLU | GSM8K | SVAMP | HUMANEVAL | SMACV2 | RWARE |
| MAPPO | 89.20% | 94.15% | 94.02% | 94.50% | – | – |
| GCN | 87.94% | 92.85% | 92.42% | 93.56% | – | – |
| CoT | 84.32% | 87.17% | 89.62% | 91.24% | 34.92% | 29.02 |
| COMPLETE GRAPH | 85.76% | 89.74% | 91.85% | 92.85% | 35.54% | 29.96 |
| LLM-BLENDER | 87.42% | 92.45% | 93.41% | 94.23% | 36.80% | 30.36 |
| AGENTPRUNE-C | 87.12% | 91.62% | 92.13% | 93.05% | 36.24% | 29.79 |
| AGENTPRUNE-L | 85.89% | 90.14% | 91.02% | 91.89% | 36.06% | 29.34 |
| AGENTPRUNE-R | 85.43% | 89.72% | 90.32% | 91.25% | 35.88% | 29.06 |
| OUR METHOD | **91.35%** | **95.86%** | **94.94%** | **95.25%** | **40.30%** | **35.40** |

*Notes. Math & Language, Complex Reasoning report accuracy (%). **SMACv2** reports mean test win-rate (%). **RWARE** reports episodic return (higher is better).*

Table 2: Comparison of accuracy, communication efficiency ($S_d$) and robustness (% Decline) on GSM8K and SMACv2. Best results are highlighted in bold.

| METHOD | ACCURACY (%) | | EFFICIENCY ($S_d$) | | ROBUSTNESS (% DECLINE) | |
|---|---|---|---|---|---|---|
| | GSM8K | SMACV2 | GSM8K | SMACV2 | GSM8K | SMACV2 |
| MAPPO | 94.15 | – | 0.68 | – | 8.21 | – |
| GCN | 92.85 | – | 0.72 | – | 8.94 | – |
| CoT | 87.17 | 34.92 | 0.85 | 0.90 | 11.23 | 14.04 |
| COMPLETE GRAPH | 89.74 | 35.54 | 0.90 | 1.00 | 10.51 | 13.14 |
| LLM-BLENDER | 92.45 | 36.80 | 0.75 | 0.83 | 9.62 | 12.02 |
| AGENTPRUNE-C | 91.62 | 36.24 | 0.72 | 0.80 | 8.71 | 10.89 |
| AGENTPRUNE-L | 90.14 | 36.06 | 0.80 | 0.88 | 9.02 | 11.27 |
| AGENTPRUNE-R | 89.72 | 35.88 | 0.81 | 0.89 | 9.86 | 12.32 |
| OUR METHOD | **95.86** | **40.30** | **0.62** | **0.68** | **7.87** | **9.84** |

Table 3: Ablation study results showing the impact of each component on GSM8K,MMLU and SMACv2 datasets. Best results are highlighted in bold.

| CONFIGURATION | GSM8K ACCURACY | MMLU ACCURACY | SMACV2 |
|---|---|---|---|
| FULL FRAMEWORK (OUR METHOD) | **95.86%** | **91.35%** | **40.30%** |
| W/O ENTROPY DECOMPOSITION | 93.21% | 89.04% | 38.66% |
| W/O DYNAMIC SPARSITY | 92.45% | 87.62% | 36.34% |
| BASELINE (COMPLETE GRAPH) | 89.74% | 85.76% | 35.54% |

tational, bandwidth, and energy constraints. Furthermore, real-world MAS exhibit highly dynamic network structures, where agents can drop out, experience network failures, or move unpredictably. An incremental update mechanism for the sparsity-aware communication structure could enhance robustness. Lastly, evaluating our approach on practical large-scale MAS benchmarks (e.g., swarm robotics, warehouse automation, traffic coordination) would provide a stronger validation of its applicability beyond synthetic settings.

## 4.3 ABLATION STUDY

To analyze the contribution of each component in our framework, we perform an ablation study by incrementally removing components such as entropy-based task decomposition and dynamic sparsity. Table 3 presents the results.

The removal of entropy-based task decomposition results in a notable decline in accuracy, with **a 2.65% drop on GSM8K, a 2.31% drop on MMLU and a 1.64% drop on SMACv2**. This highlights its critical role in achieving balanced task distribution across agents, which directly impacts overall system performance by reducing computational bottlenecks; Excluding dynamic sparsity leads to both decreased communication efficiency and a reduction in accuracy. This demonstrates that dynamic sparsity is essential for maintaining effective communication between agents by prioritizing critical links while eliminating redundant ones, thereby optimizing the trade-off between communication costs and task execution quality. These results validate the complementary contributions of all components to the overall performance of the framework.

### 4.4 DISCUSSION

The results highlight the efficacy of integrating entropy-based task decomposition, dynamic sparsity, and MDP-driven task allocation into a unified framework. The proposed approach consistently outperforms state-of-the-art baselines across multiple metrics, including accuracy and communication efficiency, while demonstrating resilience in dynamic and perturbed environments. The entropy-based task decomposition ensures balanced workload distribution among agents, while dynamic sparsity reduces redundant communication, significantly enhancing scalability and resource utilization. The integration of MDP-driven task allocation further optimizes adaptability, enabling the framework to maintain high performance under varying system conditions.

These findings underscore the potential of our method to address the challenges of large-scale multi-agent systems, particularly in scenarios with constrained communication resources or dynamically changing tasks. Future work will aim to expand the evaluation to additional datasets spanning diverse domains, enabling broader validation of the framework's generalizability. Furthermore, exploring advanced mechanisms for dynamic sparsity and task prioritization, such as real-time adjustments based on agent feedback, could further enhance the efficiency and robustness of the system.

## 5 CONCLUSION

We present a scalable and efficient framework for multi-agent systems that integrates entropy-based task decomposition, dynamic sparsity, and an MDP-driven allocator to systematically address computational, communication, and dependency complexities – balancing workload, reducing messaging, and enabling adaptive reassignment in dynamic environments. By leveraging entropy to guide task allocation, sparsity-aware communication to optimize inter-agent interactions, and reinforcement learning for decision-making, the method attains efficiency and scalability while maintaining robustness. On GSM8K, MMLU, and SMACv2, it achieves state-of-the-art accuracy – $+5.59\%$ on MMLU, $+6.12\%$ on GSM8K and $+4.76\%$ on SMACv2 – while reducing communication complexity by $28\%$, outperforming pruning-based baselines in error resilience. These results validate entropy-guided sparsity: redundant links can be pruned without sacrificing performance, reducing bandwidth and compute, and the method's sublinear communication complexity underscores suitability for large-scale, high-agent-count MAS.

While the proposed framework demonstrates strong theoretical and empirical performance, its real-world applicability is subject to certain limitations. The assumption of agent homogeneity may not hold in practical settings where computational and communication capabilities vary. Additionally, real-world multi-agent systems often face unpredictable network dynamics, task interdependencies, and large-scale scalability constraints that may impact performance. The reliance on Laplacian eigenvalue analysis introduces computational overhead, which could hinder real-time adaptation in large-scale deployments. Future work will focus on extending the framework to heterogeneous agent settings, incorporating adaptive sparsification mechanisms responsive to real-time conditions, and validating its scalability in practical applications such as autonomous robotics, decentralized AI, and smart infrastructure.

By addressing scalability, efficiency, and adaptability, our method advances the deployment of MAS in robotics, IoT, and distributed AI, paving the way for more robust, large-scale multi-agent coordination in complex and resource-constrained environments. We believe this work lays a strong foundation for future research in self-optimizing, decentralized multi-agent frameworks, bridging the gap between theoretical efficiency and real-world scalability.

## 6 ETHICS STATEMENT

**Human subjects & sensitive data.** This work does not involve human subjects, personal data, or user-generated private content. All benchmarks are publicly available and widely used in the community: GSM8K, MMLU, SVAMP, HumanEval, SMACv2, and RWARE. We follow the licenses and usage terms for each dataset and software dependency.

## 7 REPRODUCIBILITY STATEMENT

**Hyperparameters & training protocol.** Appendix D lists defaults and per-domain overrides (optimizer, batch sizes, rollout lengths/epochs, gradient clipping, MI estimator & minibatch, spectral refresh/power-iteration steps, $\lambda_2$ target, repair budget $k_r$, assignment step size and constraints). We also specify refresh cadences $K_{\mathrm{MI}}, K_\lambda$ and stopping tolerances.

**Exact algorithms.** Section 3 defines the MI-based communication priority $C_{ij} = \frac{1}{1+I(X_i;X_j)}$, adaptive threshold update equation 7, $\lambda_2$-guarded repair, entropy-balanced assignment, and the MDP allocator; Algorithm 1 gives the full loop.

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

## A  SECOND-ORDER DYNAMICS AND STABILITY

### A.1  MODEL AND LAPLACIAN MODAL DECOMPOSITION

Consider a connected, undirected, weighted graph $G = (V, E, W)$ with Laplacian $L = D - A$. The second-order consensus dynamics for positions $x(t) \in \mathbb{R}^m$ and velocities $v(t) \in \mathbb{R}^m$ (stacked agent-wise) are

$$\dot{x}(t) = v(t), \tag{10}$$
$$\dot{v}(t) = -\alpha L x(t) - \beta L v(t), \tag{11}$$

with gains $\alpha > 0$, $\beta > 0$. Let $0 = \lambda_1 < \lambda_2 \le \cdots \le \lambda_m$ be the eigenvalues of $L$, and $U^\top L U = \operatorname{diag}(\lambda_i)$ an orthonormal basis. In modal coordinates $(z, w) = (U^\top x, U^\top v)$, the dynamics decouple across Laplacian modes, and for each $\lambda > 0$ we obtain

$$\ddot{z}_\lambda(t) + \beta \lambda \dot{z}_\lambda(t) + \alpha \lambda z_\lambda(t) = 0. \tag{12}$$

The consensus subspace ($\lambda_1 = 0$) evolves as $\dot{z}_0 = w_0$, $\dot{w}_0 = 0$, so positions converge to a common trajectory governed by the average initial velocity.

### A.2  CONTINUOUS-TIME STABILITY AND RATE

For each $\lambda > 0$, the characteristic polynomial of equation 12 is $s^2 + \beta \lambda s + \alpha \lambda$, whose roots satisfy $\operatorname{Re}(s) < 0$ whenever $\alpha > 0$ and $\beta > 0$. Hence:

**Proposition A.1** (Exponential consensus for equation 11). *If $G$ is connected and $\alpha, \beta > 0$, then for any initial $(x(0), v(0))$, the disagreement component $(I - \frac{\mathbf{1}\mathbf{1}^\top}{m})x(t)$ converges to zero exponentially. A mode-wise decay rate lower bound is*

$$\rho_{\mathrm{ct}} \ge \min_{\lambda \in \{\lambda_2, \ldots, \lambda_m\}} \frac{1}{2} \left( \beta\lambda - \sqrt{(\beta\lambda)^2 - 4\alpha\lambda} \right),$$

*with the understanding that in the underdamped regime $((\beta\lambda)^2 < 4\alpha\lambda)$ the real part equals $\beta\lambda/2$.*

### A.3  SAMPLED IMPLEMENTATION AND DISCRETE-TIME STABILITY

With sampling period $T > 0$ and forward Euler discretization of equation 11:

$$x_{k+1} = x_k + T v_k, \tag{13}$$
$$v_{k+1} = v_k - T\alpha L x_k - T\beta L v_k. \tag{14}$$

For each $\lambda > 0$, the mode update is

$$\begin{bmatrix} z_{k+1} \\ w_{k+1} \end{bmatrix} = A_\lambda \begin{bmatrix} z_k \\ w_k \end{bmatrix}, \qquad A_\lambda = \begin{bmatrix} 1 & T \\ -\alpha\lambda T & 1 - \beta\lambda T \end{bmatrix}.$$

Stability requires the eigenvalues of $A_\lambda$ lie strictly inside the unit disc for all $\lambda \in \{\lambda_2, \ldots, \lambda_m\}$. Applying Jury's test to the characteristic polynomial $r^2 - (\operatorname{tr} A_\lambda)r + \det A_\lambda = 0$ yields the following sufficient, mode-uniform conditions.

**Proposition A.2** (Discrete-time stability: a simple sufficient condition). *Let $\lambda_{\max} = \lambda_m(L)$. If $\alpha > 0$, $\beta > 0$ satisfy*

$$\beta^2 < \frac{4\alpha}{\lambda_{\max}}, \tag{15}$$

*and the sampling period obeys*

$$0 < T < \frac{\beta}{\alpha}, \tag{16}$$

*then all disagreement modes of equation 14 are Schur-stable, i.e., $\rho(A_\lambda) < 1$ for every $\lambda \in [\lambda_2, \lambda_{\max}]$, and the system reaches discrete-time consensus.*

*Proof.* For $A_\lambda$ we have $\operatorname{tr} A_\lambda = 2 - \beta\lambda T$ and $\det A_\lambda = 1 - \beta\lambda T + \alpha\lambda T^2$. Jury conditions for a second-order polynomial reduce to (i) $1 - \det A_\lambda > 0$, (ii) $1 + \operatorname{tr} A_\lambda + \det A_\lambda > 0$, (iii) $1 - \operatorname{tr} A_\lambda + \det A_\lambda > 0$. Condition (iii) simplifies to $\alpha\lambda T^2 > 0$ (always true for $T > 0$). Condition (i) yields $\lambda T(\beta - \alpha T) > 0 \Rightarrow T < \beta/\alpha$. For (ii) we obtain $4 - 2\beta\lambda T + \alpha\lambda T^2 > 0$, whose minimum over $T$ on $(0, \beta/\alpha)$ is attained at $T^\star = \beta/\alpha$ with value $4 - (\beta^2\lambda)/\alpha$. Enforcing positivity at $\lambda = \lambda_{\max}$ gives equation 15. Combining (i)–(iii) gives the claim. $\square$

## A.4 INTERACTION WITH SPECTRAL SPARSIFICATION

If a sparsified Laplacian $L_S$ satisfies the standard spectral envelope $(1 - \varepsilon)L \preceq L_S \preceq (1 + \varepsilon)L$, then

$$(1 - \varepsilon)\lambda_2(L) \leq \lambda_2(L_S) \leq \lambda_{\max}(L_S) \leq (1 + \varepsilon)\lambda_{\max}(L).$$

Thus any gain/sampling choices $(\alpha, \beta, T)$ that satisfy equation 15 and equation 16 with $\lambda_{\max}$ replaced by $(1 + \varepsilon)\lambda_{\max}(L)$ remain sufficient for discrete-time stability under sparsification; alternatively, the sampled-data Nyquist condition $T < \frac{1}{2\sqrt{\alpha\lambda_{max}}}$ (equation 27 in Appendix B) also suffices. In this paper we use the Jury bounds equation 15 and equation 16 by default, and the continuous-time rate in A.1 degrades by at most a factor depending on $(1 - \varepsilon)\lambda_2(L)$.

# B STABILITY ANALYSIS OF SAMPLED-DATA CONTROL

## B.1 SYSTEM MODEL AND STABILITY CONDITION

Multi-agent systems operating under communication constraints rely on *sampled-data control* (Gao et al., 2009), where information is exchanged at discrete time instants $\{t_k\}_{k=0}^{\infty}$ with a fixed sampling period $T = t_{k+1} - t_k$. The system dynamics under sampled-data control are described as follows:

$$\dot{x}_i(t) = v_i(t), \tag{17}$$

$$\dot{v}_i(t) = \alpha \sum_{j \in \mathcal{N}_i} a_{ij}\big(x_j(t_k) - x_i(t_k)\big) + \beta \sum_{j \in \mathcal{N}_i} a_{ij}\big(v_j(t_k) - v_i(t_k)\big), \quad t \in [t_k, t_{k+1}). \tag{18}$$

This discretization introduces a trade-off: smaller $T$ improves synchronization but increases communication overhead, whereas larger $T$ reduces communication cost but may cause instability due to delayed state updates. We now establish a sufficient bound on $T$ for stability.

The bound below is alternative to the Jury-based sufficient conditions in Appendix A.3; both are valid, with different constants and assumptions.

**Assumption B.1.** The communication graph is undirected and connected with weighted adjacency matrix $A = [a_{ij}]$, and the Laplacian matrix $L$ has eigenvalues $\lambda_1 = 0 < \lambda_2 \leq \cdots \leq \lambda_{\max}$.

**Definition B.2.** The *Nyquist sampling criterion* states that for a continuous-time system with maximum eigenfrequency $\sqrt{\alpha\lambda_{\max}}$, the sampling period $T$ must satisfy:

$$T < \frac{1}{2\sqrt{\alpha\lambda_{\max}}} \tag{19}$$

to prevent aliasing and ensure stability (Desoer & Wang, 1980).

**Lemma B.3.** *For the continuous-time model dynamics, the characteristic polynomial is:*

$$s^2 + \beta\lambda_i s + \alpha\lambda_i = 0, \quad i = 1, 2, \ldots, N. \tag{20}$$

*The roots of this equation determine the stability of the system.*

*Proof.* By defining the state vector $z(t) = [x(t)^\top, v(t)^\top]^\top$, the system can be rewritten as:

$$\dot{z}(t) = Az(t), \tag{21}$$

where the system matrix is:

$$A = \begin{bmatrix} 0 & I \\ -\alpha L & -\beta L \end{bmatrix}. \tag{22}$$

The eigenvalues of $A$ govern the system's stability. Solving $\det(sI - A) = 0$, we obtain the characteristic equation given in B.3. $\qquad\square$

**Theorem B.4.** *An alternative sufficient stability condition is obtained if the sampling period satisfies:*

$$T < \frac{1}{2\sqrt{\alpha\lambda_{\max}}}. \tag{23}$$

*Proof.* Using the zero-order hold (ZOH) discretization, the discrete-time state transition is:

$$z(t_{k+1}) = e^{AT} z(t_k). \tag{24}$$

For stability, all eigenvalues of $e^{AT}$ must lie within the unit disk (from the model analysis in Appendix A.3). The dominant eigenfrequency of the system is $\sqrt{\alpha\lambda_{\max}}$, and by Nyquist's sampling criterion (Desoer & Wang, 1980), the Nyquist frequency is:

$$\omega_N = \frac{\pi}{T}. \tag{25}$$

To prevent aliasing and instability, the system's maximum eigenfrequency must be below Nyquist's limit:

$$\sqrt{\alpha\lambda_{\max}} < \frac{\pi}{T}. \tag{26}$$

Rearranging the inequality yields the sampling constraint:

$$T < \frac{1}{2\sqrt{\alpha\lambda_{\max}}}. \tag{27}$$

$\square$

**Corollary B.5.** *A smaller $T$ enhances synchronization and stability but increases communication overhead. Conversely, a larger $T$ reduces communication costs but increases the risk of instability. The bound in B.4 provides a sufficient stability-communication trade-off.*

The derived constraint ensures synchronization updates remain within stability limits, balancing communication efficiency and robustness in multi-agent systems. This result is particularly relevant for real-world systems constrained by bandwidth (Yu et al., 2024) and computational limitations (Tripathi et al., 2021).

## C   THEORETICAL JUSTIFICATION OF ENTROPY-GUIDED SPARSITY

### C.1   ENTROPY-GUIDED SPARSITY PRESERVES KEY COMMUNICATION LINKS

**Theorem C.1.** *(**Connectivity via Spectral-Aware Repair after Entropy Thresholding**)*
*Let $G = (V, E, W)$ be the original multi-agent communication graph, where $W$ is the weighted communication matrix. Define the entropy-based pruning threshold $H_{thresh}$, and retain only edges $(i, j) \in E$ satisfying:*

$$H(e_{ij}) \geq H_{thresh}, \tag{28}$$

*where $H(e_{ij})$ is the Shannon entropy of the information exchanged between agents $i$ and $j$. Then, the sparsified graph $G_S = (V, E_S)$ satisfies:*

1. *(**Connectivity via Repair**) Let $E_{thresh}$ be the edge set after entropy-thresholding. Apply the spectral-aware repair step from Appendix D.5 that adds up to $k_r$ edges using a Fiedler/effective-resistance heuristic until $\lambda_2(L_S) \geq \lambda_{target} > 0$. If the target is reached, the resulting graph $G_S = (V, E_S)$ with $E_S = E_{thresh} \cup E_{repair}$ is connected, equivalently, $\lambda_2(L_S) \geq \lambda_{target}$.*
2. *(**Edge count**) Let $k_{add} \leq k_r$ be the number of edges added by repair. Then*

$$|E_S| = |E_{thresh}| + k_{add}, \quad k_{add} \leq k_r. \tag{29}$$

*If each node keeps at most $k$ neighbors in the threshold step, then $|E_{thresh}| \leq \frac{km}{2}$ and hence $|E_S| \leq \frac{km}{2} + k_r$. Without a degree cap, $|E_{thresh}| = S_d \cdot |E|$ with $S_d \in (0, 1]$, and the same additive $k_r$ term applied.*

*Proof.* Entropy $H(e_{ij})$ quantifies the uncertainty and informativeness of a communication link. High-entropy edges introduce novel information, while low-entropy edges indicate redundant communication. Entropy-thresholding yields a candidate subgraph; connectivity is then enforced by the spectral-aware repair routine from Appendix D.5, which adds edges until $\lambda_2(L_S) \geq \lambda_{target}$. Under C.3, the resulting $G_S$ also satisfies the $(1 \pm \varepsilon)$ spectral envelope stated in C.4. $\square$

## C.2 Relationship to Mutual Information and Redundancy Reduction

**Proposition C.2.** *(Mutual Information-Based Sparsity Reduction)*
*Let $I(A_i; A_j)$ denote the mutual information between two agents $i$ and $j$. If entropy-based sparsification removes edges where:*

$$I(A_i; A_j) \geq I_{thresh}, \tag{30}$$

*then the expected loss in task performance $\mathbb{E}[\Delta J]$ satisfies:*

$$\mathbb{E}[\Delta J] \leq O\left(\frac{|E| - |E_S|}{|E|}\right). \tag{31}$$

*Under a Lipschitz task-return model, we hold:*

$$\mathbb{E}[\Delta J] \leq L_J \cdot \frac{|E| - |E_S|}{|E|}, \tag{32}$$

*i.e., degradation scales with the fraction of pruned edges.*

*Proof.* Mutual information $I(A_i; A_j)$ quantifies how much information two agents already share. High $I(A_i; A_j)$ implies redundancy, meaning that communication between $i$ and $j$ does not contribute significant new knowledge. By pruning edges with high $I(A_i; A_j)$, we minimize redundancy while preserving task-critical exchanges. A first-order Taylor expansion confirms that expected task degradation is upper-bounded by the fraction of edges removed. □

## C.3 Spectral Sparsification and Task Performance Preservation

**Assumption C.3.** (**Edge Selection & Degree**)
The repair step from Appendix D.5 adds back up to $k_r$ edges chosen by an approximate effective-resistance or Fiedler-leverage heuristic, and the resulting subgraph has bounded maximum degree. Under these conditions, $G_S$ is a $(1 \pm \varepsilon)$ spectral sparsifier of $G$ for some $\varepsilon \in (0, 1)$.

*Remark.* Classical results imply the existence of $(1 \pm \varepsilon)$ spectral sparsifiers with $O\left(\frac{m \log m}{\varepsilon^2}\right)$ edges (Spielman & Srivastava, 2008); our threshold with repair routine approximates this behavior heuristically.

**Theorem C.4.** *(Spectral Graph Sparsification under Entropy-Based Pruning)*
*Let $L$ be the Laplacian matrix of the original graph $G$ and $L_S$ the Laplacian of the sparsified graph $G_S$. Then, for sparsification parameter $\epsilon > 0$, we have:*

$$(1 - \epsilon)L \preceq L_S \preceq (1 + \epsilon)L. \tag{33}$$

*Proof.* From spectral sparsification, under C.3, standard sparsification results imply that for all $x \in \mathbb{R}^m$,

$$(1 - \varepsilon)x^\top L x \leq x^\top L_S x \leq (1 + \varepsilon)x^\top L x, \tag{34}$$

which is equivalent to $(1 - \varepsilon)L \preceq L_S \preceq (1 + \varepsilon)L$. If edges are removed in a way that preserves the dominant eigenvalues of $L$, the communication structure remains stable. Entropy-guided pruning selectively retains edges with high information flow, ensuring the sparsified Laplacian $L_S$ approximates the full graph Laplacian within error bounds of $\epsilon$. This guarantees that multi-agent coordination properties are maintained. □

**Corollary C.5.** *If entropy-guided sparsification satisfies C.4, the total performance degradation $\Delta J$ is bounded by:*

$$\Delta J = O\left(\frac{|E| - |E_S|}{|E|}\right). \tag{35}$$

*Under a Lipschitz task-return model with constant $L_J$,*

$$\Delta J \leq L_J \cdot \frac{|E| - |E_S|}{|E|} \tag{36}$$

*Proof.* Directly follows from C.2 and C.4. □

## D LEARNING AND IMPLEMENTATION DETAILS

### D.1 MDP FORMALIZATION

We cast adaptive coordination as an MDP $(\mathcal{S}, \mathcal{A}, P, R, \gamma)$. At step $t$:

- **State** $s_t$ summarizes system load and graph statistics, e.g., (utilization vector $u_t$, queue/latency $q_t$, $\lambda_2(L_{S,t})$, $\widehat{\lambda}_{\max}(L_{S,t})$, $S_{d,t}$, task mix $\pi_t$, MI summary $\bar{I}_t$).
- **Action** $a_t$ adjusts (i) communication sparsity via a threshold $\tau_t$ (or a budget $b_t$) and optional repair steps to meet $\lambda_2(L_{S,t}) \geq \lambda_{\text{target}}$, and (ii) task weights/assignments via a step on the entropy objective (Eq. (3) in the main text).
- **Reward** $R(s_t, a_t)$ trades performance against messaging:

$$R_t = J_t - \lambda_{\text{msg}} \cdot \text{Msgs}_t - \lambda_{\text{viol}} \cdot \left[ \lambda_{\text{target}} - \lambda_2(L_{S,t}) \right]_+,$$

where $J_t$ is task return/accuracy, $\text{Msgs}_t$ is per-step message count, and $[x]_+ = \max\{0, x\}$ penalizes spectral violations.

The transition $P$ is induced by the environment/task generator and the graph update.

### D.2 Q-LEARNING VARIANT (VALUE-BASED CONTROL)

We use tabular or function-approximation Q-learning for discrete action grids (e.g., $\tau \in \{\tau_1, \ldots, \tau_K\}$, step-sizes for assignment updates) (Watkins & Dayan, 1992).

---

**Algorithm 2** Adaptive Sparse Communication & Entropy-Balanced Tasking (Q-learning)

1: **Input:** discount $\gamma$, learning rate $\eta$, exploration $\epsilon$, target $\lambda_{\text{target}}$ (threshold adaptation uses $\kappa$ as in equation 7 of the main text)
2: Initialize $Q(s, a) \leftarrow 0$ for all $(s, a)$ (or NN weights), initial graph $L_{S,0}$
3: **for** each episode **do**
4:     Reset environment; observe $s_0$
5:     **for** $t = 0, 1, \ldots, T_{\max} - 1$ **do**
6:         Select $a_t$ via $\epsilon$-greedy over $Q(s_t, \cdot)$
7:         Apply $a_t$: update sparsity threshold $\tau_t$ (and repair to enforce $\lambda_2(L_{S,t}) \geq \lambda_{\text{target}}$); take one projected step on the entropy objective for $P_t$
8:         Execute tasks; observe reward $R_t$ and next state $s_{t+1}$
9:         $Q(s_t, a_t) \leftarrow Q(s_t, a_t) + \eta \big( R_t + \gamma \max_{a'} Q(s_{t+1}, a') - Q(s_t, a_t) \big)$
10:     **end for**
11: **end for**

---

For continuous controls (e.g., real-valued $\tau$), use tile coding or a small MLP. We refresh MI statistics every $K_{\text{MI}}$ steps and estimate $\lambda_2$ with a few power iterations (reusing Krylov vectors when possible).

### D.3 POLICY-GRADIENT VARIANT (ACTOR–CRITIC / PPO)

For smoother control over continuous actions, an actor $\pi_\theta(a|s)$ and critic $V_\phi(s)$ are trained with clipped PPO. At each step we (i) sample $a_t$, (ii) apply sparsity/assignment updates, (iii) compute $R_t$, and (iv) update $\theta, \phi$ using standard PPO objectives with advantage estimates $\hat{A}_t$ (GAE). The spectral penalty integrates naturally into the reward.

### D.4 ASSIGNMENT SOLVER DETAILS

We implement Eq. (3) in the main text with projected mirror descent on the simplex: $P \leftarrow \text{Proj}_{\mathcal{C}}\big(P + \eta_P(\nabla H_{\text{row}} + \lambda \nabla H_{\text{col}})\big)$, where $\mathcal{C} = \{P : \sum_j P_{ij} \leq c_i, \sum_i P_{ij} = 1, P \in [0, 1]\}$. A few inner iterations (1–3) per environment step suffice; capacity violations are clipped and renormalized per column.

### D.5 SPECTRAL-AWARE SPARSITY UPDATE

We set the redundancy score $r_{ij} := C_{ij} = \frac{1}{(1+I(X_i;X_j))}$ (equation 6 in the main text). We form a candidate active set by thresholding $r_{ij}$ at $\tau_t$, then apply a lightweight repair: add back up to $k$ edges selected by an approximate effective-resistance or Fiedler-vector leverage heuristic until $\lambda_2(L_{S,t}) \geq \lambda_{\text{target}}$ (or budget is met). This preserves global connectivity while keeping $S_d$ low.

### D.6 COMPUTATION & COMMUNICATION COMPLEXITY (PER ENVIRONMENT STEP)

Let $m = |V|$ (agents), $|E|$ the current edges, and $n$ the tasks. Communication-side costs scale with $|E|$ ($\approx S_d \cdot m^2$ in dense graphs or $\approx m \cdot k$ if each node keeps $k$ neighbors), while assignment-side costs scale with $m \cdot n$.

- MI refresh (mini-batch, $b$ samples): $\tilde{O}(b|E|)$ for simple moment-based estimators; MINE-style estimators add a small NN update.
- $\lambda_2$ estimate (power iteration, $k$ steps): $O(k|E|)$ sparse matvecs.
- Repair (add-back $k_r$ edges): $O(k_r \log m)$ if using priority queues / precomputed scores.
- Assignment update (mirror descent, $t_P$ inner steps): $O(t_P\, mn)$.

In practice, MI and spectral estimates are amortized by refreshing every $K_{\text{MI}}/K_\lambda$ steps.

### D.7 TRAINING PROTOCOL AND DEFAULT

**Randomness & evaluation.** We report mean over 5/10 seeds runs (per domain: GSM8K/MMLU/SVAMP/HumanEval: 5, SMACv2/RWARE: 10). Dataset splits are fixed where applicable. Checkpoints are selected by best validation score; test is run once per seed.

**General RL settings.** Discount factor $\gamma = 0.99$; optimizer Adam ($\beta_1 = 0.9, \beta_2 = 0.999$); learning-rate $\eta \in [1e-4, 1e-3]$ unless overridden; gradient-norm clip 0.5.

**Q-learning (value-based).** $\varepsilon$-greedy annealed $0.2 \rightarrow 0.02$ over $3 \cdot 10^5$ env steps; replay buffer $1e6$; batch size 256; target-net update every $1e4$ steps; Q-net MLP $[256, 256]$ ReLU.

**PPO (actor-critic).** Clip 0.2, GAE $\lambda = 0.95$, entropy bonus $1e-3$; rollout length 2048, PPO epochs 4, minibatch 64, policy/value MLP $[256, 256]$ ReLU; value-loss coeff 0.5.

**Communication-cost terms.** Messaging penalty $\lambda_{\text{msg}} \in [0.01, 0.1]$ (domain-specific); violation penalty $\lambda_{\text{viol}} = 1.0$.

**MI estimation (edge priority).** Default estimator: moment-based (Gaussian plug-in); MI mini-batch $b = 512$; feature window $W = 64$; refresh every $K_{\text{MI}} = 10$ steps.

**Spectral estimates & repair.** Power iteration $k = 20$ per refresh; refresh every $K_\lambda = 5$; repair budget $k_r \in [\lfloor 0.01|E| \rfloor, \lfloor 0.05|E| \rfloor]$ with effective-resistance/Fiedler heuristic until $\lambda_2(L_S) \geq \lambda_{\text{target}}$.

## E  LLM USAGE DISCLOSURE

**Writing assistance.** We used **ChatGPT-5** for grammar polishing only (no content generation); all technical content was drafted and verified by the authors.

