# OpenReview forum: "Spectral-Aware Sparse Communication and Entropy-Balanced Tasking in Multi-Agent Systems"
_ICLR.cc/2026/Conference — Submitted to ICLR 2026_

### Official Review · Reviewer_owk9 · 2025-10-26

**Soundness:** 1
**Presentation:** 1
**Contribution:** 2
**Rating:** 4
**Confidence:** 2

**Summary:**

This paper proposed a communicaiton-efficient framework for multi-agent systems. Redundent links are removed based on spectral graph theory. Workloads among agents are coordinated and balanced based on entropy regularization. An MDP is employed to adaptively adjust threshold for pruning. Experiments show that their algorithm achieve better accuracy on various tasks.

**Strengths:**

Their motivation is clear and strong. Reducing redundency in communication is an important problem in designing MAS. Incorporating spectral graph theory and entropy in their approach is reasonable.

**Weaknesses:**

(1) This paper is hard to read and understand. For example, it is not made clear what $X_i$ in Equation(6) is. Is it hidden states in LLM? There are also no examples showing how their methods are implemented in practice.

(2) Several important parts of their experiment setup are missing. For example, what is their LLM backend.

(3) Only accuracy is reported. No costs or efficiency are shown.

(4) Definition of communication priority matrix is heurisitic. It would be better if more explanation and ablations can be provided.

**Questions:**

(1) How is $X_i$ in Equation(6) formally defined?

(2) What is LLM backend used for LLM agents in experiments of this paper?

(3) Is there any ablations on different choices for communication priority matrix?

(4) Is there any ablations which isolates the MDP part?

---

### Official Review · Reviewer_dcnB · 2025-11-01

**Soundness:** 2
**Presentation:** 2
**Contribution:** 2
**Rating:** 2
**Confidence:** 3

**Summary:**

The paper tells a compelling, principled story; information-aware sparsification with a spectral guard plus entropy-balanced tasking, tied together by an adaptive controller and backs it with clean theory and promising empirical gains. However, the experimental layer does not yet substantiate practical communication savings for LLM-based Multi Agent Systems (no bytes/tokens/latency/energy/cost), Mutual Information estimation and global spectral operations are under-specified and likely costly, and LLM/task-setup details are insufficient for reproducibility or apples-to-apples comparisons.

**Strengths:**

The positives of the paper can be summarised as follows,

- Cohesive, systems-minded formulation. The paper unifies the link pruning via an information-theoretic priority with $\lambda_{2}$ (algebraic connectivity) guard with repair, entropy-regularized task assignment under capacity constraints, and an MDP controller to adapt thresholds/repairs. Overall, this is a neat e2e story for scaling multi-agent systems under bandwidth/latency constraints.
- Concrete spectral guarantees and stability analysis. The method enforces a spectral envelope and a target as in $\lambda_{\text{target}}$ which is used to derive the continuous/discrete stability via standard Laplacian-mode and Jury/Nyquist bounds; appendices connect how sparsification perturbs rates/conditions.
- Measured comms/robustness alongside accuracy. Results report accuracy/win-rate plus edge ratio and robustness under 30% random edge drops; the method is both most accurate and most frugal on GSM8K/MMLU/SMACv2 and is least affected by edge removal which is a nice plus!
- Clear limitations section. The paper is upfront about homogeneous-agent assumptions, eigen-analysis overheads, and the gap to real, large-scale MAS deployments.

**Weaknesses:**

The negatives of the paper can be summarised as follows,

- Applicability to LLM benchmarks is underspecified. How "agents" are instantiated for GSM8K/MMLU/HumanEval is rather unclear (backbone LLMs, number of agents, prompt/temperature/budget parity, tool use, per-turn token limits). Without these details, accuracy improvements over "Complete Graph", CoT, or LLM-Blender are hard to interpret or reproduce...
- Communication cost is proxied by $\|E\|$, not bytes/tokens as per usual. The paper equates per-iteration cost to active edge count, but message size dominates in LLM settings (context windows, chain-of-thought, function-call payloads, thinking budget - if any). No "bytes on the wire", latency, or energy measurements are reported. This weakens the "28% reduction" claim as a real systems improvement for LLM agents.
- Mutual Information-driven pruning can be quite expensive and is under-specified within the paper. Computing/refreshing pairwise mutual information for all agent pairs is $O(m^2)$ and may dominate savings; the estimator, windowing, and sample complexity are not specified, and the effect of estimation noise on the spectral guard/repairs is not studied.
- Robustness protocol is unclear. For the 30% edge-drop test, are drops i.i.d. at inference with/without allowing repairs? Are all methods equally allowed to adapt? If your controller adapts but baselines are frozen, the comparison may favor your approach.
- Reproducibility is limited. While Algorithm 1 and hyperparameters are listed, the paper does not state that code/splits/prompts will be released, the reason why this is the case is unclear.

**Questions:**

My questions to the authors are,

- LLM agent setup. How many agents, which LLM(s), decoding settings (temperature/top-p), tool use, and per-turn token budgets? Are all methods matched on cost (tokens, wall-clock, calls)? Please provide a table of LLM/backbone, prompts, and budgets for GSM8K/MMLU/HumanEval.
- MI estimation details. Which estimator, window length, and sample complexity? How sensitive is pruning to MI noise, and how often is MI refreshed relative to task horizon? Ideally, provide the necessary ablations.
- Entropy vs specialization. Do you support heterogeneous capacities/skills (per-agent weights or competence priors) in the objective? Show results where agents differ in compute/latency/accuracy.
- Robustness protocol. For the 30% edge-drop test, confirm whether methods can adapt/repair during drops. If yes, allow the same adaptation for baselines (or freeze all) and report both settings.
- Reproducibility concerns, will you release code, prompts, splits, and SMACv2 configs to reproduce Tables 1–3? If not, please include an anonymized artifact.

---

### Official Review · Reviewer_t9k2 · 2025-11-01

**Soundness:** 2
**Presentation:** 2
**Contribution:** 2
**Rating:** 2
**Confidence:** 3

**Summary:**

This paper introduces a unified framework for multi-agent systems designed to enhance scalability and efficiency. The method combines three core components: (1) a spectral-aware communication module that prunes redundant links based on an information-theoretic priority while enforcing network connectivity via a Laplacian eigenvalue (λ2) guard; (2) an entropy-regularized tasking module to ensure balanced workload distribution; and (3) an MDP-based allocator that dynamically adapts system parameters. The authors provide theoretical analysis for stability and convergence and demonstrate empirical improvements in performance, efficiency, and robustness on tasks including GSM8K, MMLU, and SMACv2.

**Strengths:**

1. The core idea of using an information-theoretic metric (mutual information) to prune redundant communication links is intuitive and well-motivated.
2. The framework's ambition to create a unified solution that jointly optimizes communication and dynamic adaptation is commendable.

**Weaknesses:**

1. My main concern is that the method description is high-level and lacks clarity, making it difficult to understand how the components are implemented. The paper describes communication pruning, entropy-based tasking, and an MDP allocator as high-level concepts, but the particular implementation are not explained.
2. The authors mention that the dense communication brings network congestion and heavy communication overhead. Nevertheless, the proposed pruning mechanism requires computing pairwise mutual information across all agents. This process seems inherently centralized and communication-intensive, as it requires gathering information from agent pairs to a central point for computation. This appears to trade one form of communication overhead (dense task-related messaging) for another (dense feature-gathering for MI calculation). The paper needs to explicitly address this trade-off and justify how the overall system achieves a net reduction in communication, especially in a decentralized setting.

**Questions:**

1. Could the authors provide a precise definition of the agent features $X_i$ used to compute the mutual information? Is it fixed or is it time-relevant?
2. Is the communication topology static or dynamic within a single episode? Is the topology fixed for an episode after an initial optimization phase, or is it re-evaluated periodically during the episode?
3. The reported win-rates on SMACv2 appear low compared to state-of-the-art MARL algorithms. What are the evaluation scheme for this benchmark?

---

### Official Review · Reviewer_31aZ · 2025-11-04

**Soundness:** 1
**Presentation:** 1
**Contribution:** 1
**Rating:** 0
**Confidence:** 4

**Summary:**

The work aims to solve three primary challenges in LLM-powered multi-agent systems (MAS):
1.	High Communication Overhead: As MAS scales, dense or fully-connected communication graphs become untenable, leading to quadratic costs in messaging, network congestion, and high bandwidth and energy consumption.
2.	Imbalanced Tasking and Bottlenecks: The heterogeneity of agents, combined with dynamically generated tasks, often results in "imbalanced tasking" or "workload skew." This creates computational bottlenecks, inefficient resource distribution, and suboptimal performance.
3.	Non-Stationarity and Complexity: These issues are compounded in LLM-driven systems. Unlike traditional MAS with predefined tasks, these workloads are "non-stationary." This "increased task uncertainty" introduces greater complexity in managing dependencies and balancing computational loads in real-time.

**Strengths:**

The challenges this paper aims to solve sound interesting.

**Weaknesses:**

I have severe concerns regarding the scientific validity and coherence of this manuscript, and suspect that this paper is AI-generated or generated by the so-called “AI scientist”. The following are some evidences I found in the paper.

1.	Missing and Inappropriate Literature Review: The manuscript conspicuously lacks a "Related Work" section, which is a critical omission for any scholarly submission. Furthermore, the citation practices are highly questionable. For example, on line 269, the discussion of "reinforcement learning methods" is supported by a citation that is a survey on Safe Reinforcement Learning [1]. However, the surrounding context and the paper's focus bear no relevance to safety, making this citation entirely inappropriate.

2.	Lack of Clarity in Methodology: The paper is exceptionally difficult to read and understand. The methodology section fails to provide a clear explanation of the proposed approach. Specifically, the provided pseudocode does not adequately illustrate how the method is integrated with or leverages LLM-powered Multi-Agent Systems (MAS).

3.	Ridiculous Baseline Selection: The choice of baselines is deeply problematic. First, it includes methods that are significantly outdated (e.g., work dating back to 2001). Second, the authors claim to have evaluated LLM-based baselines on environments such as SMACv2 and RWARE. This appears to be fundamentally flawed, as these are state-vector-based environments, and it is unclear how a text-based LLM method could be meaningfully applied or benchmarked in this context.

4.	Inconsistent and Illogical Experimental Results: The results presented in Tables 1 and 2 contain critical inconsistencies. The authors report that MAPPO—a model-free, non-LLM, MARL algorithm—was evaluated on LLM-specific benchmarks (GSM8K and MMLU). Conversely, they fail to report MAPPO's results on standard MARL benchmarks (SMACv2 and RWARE), where it should be applicable. This is illogical. MAPPO is not a method for fine-tuning large models, and its evaluation on text-based benchmarks like GSM8K and MMLU seems to represent a categorical misunderstanding or misapplication of the algorithm.


[1] Gu, S., Yang, L., Du, Y., Chen, G., Walter, F., Wang, J., & Knoll, A. (2024). A review of safe reinforcement learning: Methods, theories and applications. IEEE Transactions on Pattern Analysis and Machine Intelligence.

**Questions:**

N/A

---

### Meta-Review · Area_Chair_N8Qe · 2026-01-06

**Summary:**

All reviewers unanimously recommended to reject the paper based on a variety of serious concerns, including lack of technical clarity, illogical experiments, lack of literature review, etc.

**Reviewer Concerns:**

No rebuttal

**Reviewer Scores:**

Unchanged

---

### Decision · Program_Chairs · 2026-01-26

Reject